

# Mesospheric gravity waves and their sources at the South Pole

Dhvanit Mehta[1], Andrew J. Gerrard[1], Yusuke Ebihara[2], Allan T. Weatherwax[3], and Louis J. Lanzerotti[1]

[1]Center for Solar-Terrestrial Research, New Jersey Institute of Technology, 323 Martin Luther King Jr. Boulevard, 101 Tiernan Hall, Newark, NJ 07102-1982, USA.
[2]Research Institute for Sustainable Humanosphere, Kyoto University, Gokasho, Uji City, Kyoto Prefecture, Japan.
[3]Merrimack College, 315 Turnpike St, North Andover, MA 01845, USA.

*Correspondence to:* D. Mehta
(dm36@njit.edu)

**Abstract.** The sourcing locations and mechanisms for short period, long vertical wavelength upward-propagating gravity waves at high polar latitudes remain largely unknown. Using all-sky imager data from the Amundsen-Scott South Pole Station we determine the spatial and temporal characteristics of 94 observed small-scale waves in three austral winter months in 2003 and 2004. These data, together with background atmospheres from synoptic and/or climatological empirical models, are used
to model gravity wave propagation from the polar mesosphere to each wave's source using a ray-tracing model. Our results provide a compelling case that a significant proportion of the observed waves are launched in several discrete layers in the tropopause and/or stratosphere. Analyses of synoptic geopotentials and temperatures indicate that wave formation is a result of baroclinic instability processes in the stratosphere and the interaction of planetary waves with the background wind fields in the tropopause. These results are significant for defining the influences of the polar vortex on the production of these small-scale,
upward propagating gravity waves at the highest polar latitudes.

## 1   Introduction

The breaking and induced drag caused by atmospheric gravity waves plays an important role in the dynamics of the mesosphere-lower thermosphere (MLT) region (Fritts and Alexander, 2003). The impacts of such wave breaking is felt on a climatological scale; e.g. gravity waves fundamentally drive a meridional circulation resulting in a cool summer mesopause and warm winter
mesopause (Meriwether and Gerrard, 2004). On the synoptic scale the effects of gravity waves can be seen in the localized destruction of mesospheric clouds (Gerrard et al., 2002, 2004), mesospheric fronts/bores (Brown et al., 2004), and localized wave ducting (Li et al., 2011). As such, because of their significance to the dynamics of the middle atmosphere gravity waves have been a focus of active and ongoing research, particularly at high latitudes. However, observations at high latitudes are difficult to obtain due to experimental logistics. This is even more of an issue in the Antarctic region, where few manned stations exist
to operate gravity wave instrumentation during the austral winter.

Of particular interest to this study is the determination of high latitude gravity wave source regions. Many studies have investigated the excitation of gravity waves in the lower atmosphere (Sato and Yoshiki, 2008; Gerrard et al., 2011; Moffat-Griffin et al., 2011), directly in the MLT region from auroral heating (Oyama and Watkins, 2012), and on the characteristics





and seasonal variation of gravity waves in the polar MLT region (Nielsen et al., 2012; Suzuki et al., 2011). While the excitation and propagation of gravity waves during disturbed conditions, such as during sudden stratospheric warmings and stratospheric temperature enhancements (Meriwether and Gerrard, 2004), have been investigated by Wang and Alexander (2009); Yamashita et al. (2010); Gerrard et al. (2011), there is a significant gap in understanding of wave generation during quiet conditions or

from a climatological or quasi-climatological perspective.

One dominant gravity wave source region known to occur at polar latitudes is the polar vortex (Duck et al., 1998; Whiteway and Duck, 1999). Displacement of the polar vortex away from its mean position over a pole can result in a vertically slanted, tilted wind structure that can give rise to baroclinic instabilities (Tanaka and Tokinaga, 2002). These instabilities have been studied as a generating mechanism for larger-scale (on the order of several hundred kilometer) gravity waves through extensive

modeling (Fairlie et al., 1990; O'sullivan and Dunkerton, 1995; Plougonven and Snyder, 2007; Lin and Zhang, 2008) and observational (Guest et al., 2000; Plougonven et al., 2003; Lane et al., 2004; Gerrard et al., 2011) efforts, but to date their status as a source of small scale gravity waves ($< 100$ km) has not been investigated.

In this paper we show gravity wave observations from South Pole Station, Antarctica (hereafter SPA) from a dataset previously presented in Suzuki et al. (2011). We then model the propagation of the observed waves from their site of observation

above SPA to their lower altitude sources using ray-tracing techniques. We then analyze the potential source regions of the waves using lower atmospheric analyses. In Section 2 we present our gravity wave observations. In Section 3, the results of our ray-tracing model runs are presented, with results showing stratified layers of gravity wave sources in a region around the SPA site tightly restricted in latitude. In Section 4, we show lower atmospheric analyses that support the results of our modeling efforts and our interpretation of baroclinic instability as the primary mechanism of gravity wave generation by the polar vortex.

Finally, we present conclusions in Section 5, with a discussion as to the challenges and limitations of our investigation.

## 2    Gravity wave observations

For this study we utilized data obtained from a multi-wavelength all-sky imager located at SPA, originally constructed and operated by the National Institute of Polar Research (NIPR), and now operated by the Research Institute for Sustainable Humanosphere (RISH) of Kyoto University, Japan, in collaboration with NJIT (Ejiri et al., 1999; Suzuki et al., 2011). The

imager consists of a fish-eye lens providing $180°$ field of view (Nikkor $f = 6$ mm, F1.4), a rotating filter wheel with five filters (427.8 nm, 557.7 nm, 630.0 nm, 589.0 nm, 486.1 nm) for both auroral and airglow observations, and a temperature controlled CCD camera with 512 x 512 pixel resolution. Due to its location at SPA, the system is able to operate more or less continuously during the austral winter period, between April and August barring periods where the moon is at high elevation angle. In this paper we chiefly focused on the green line OI (557.7 nm) and Na (589.0 nm) airglow filters. For data shown from 2003 and

2004, Na images have 64 sec exposure times and are taken roughly 100 sec apart, while green line images are taken with 8 sec exposures, also at 100 sec sampling rate.

Gravity wave observations have previously been reported with this instrument using its Na airglow filter for the 2003-2005 austral winters by Suzuki et al. (2011), providing a climatology of waves observed at $\sim95$ km for both larger-scale "band"





events as well as smaller scale "ripple" events that are commonly thought to be localized convective or dynamical instability processes. For our own analysis, we used a portion of this data set covering July 2003, August 2003, and August 2004 as these periods showed the highest continuous Na airglow observations with minimal contamination by auroral emissions. Note that while the 589.3-nm emission is generally not sensitive to auroral contamination, we nonetheless found the presence of auroral

emissions in our image data, likely as a result of spectral leakage due to complications with the filter. While this contamination was only problematic during periods where the auroral emissions were particularly bright, its persistence throughout the data set meant we were forced to compare our images with roughly simultaneous green-line 557.7 nm filter images taken from the same instrument. This allowed us a greater accuracy in differentiating between auroral processes and gravity wave signatures in our Na images and allowed us to observe gravity waves even in conditions where portions of the image were contaminated.

Prior to analyzing images for the signatures of gravity waves, it was necessary to apply a number of post-processing techniques to the data. First, to correct for distortion of the image as a result of the fish-eye lens, images were unwarped using the technique described in Garcia et al. (1997) into geographic coordinates from the original "warped" image coordinate frame. Next, the resultant images were time-differenced in order to heighten image contrast and make it possible to identify gravity wave structure in the fairly faint airglow emission. Finally, the images were band pass filtered. While many studies using newer

imager systems eschew time-differencing due to the potential introduction of artifacts, it was necessary in our analysis due to the faintness of the emission, as well as the significant difference in contrast between airglow and auroral contamination any time contamination was present. Once the images were fully processed, images were inspected for the presence of gravity waves and their observed horizontal wavelengths, periods, and propagation directions were measured and recorded.

From the 38 days of available data during July 2003, August 2003, and August 2004, we observed 94 total wave events.

Examples are shown in Figures 1 and 2. In Figure 1, for August 6, 2004, a gravity wave is seen propagating southward at 207° with $\lambda_h$ = 17 km and $T_{obs}$ = 7.9 min beginning around 11:37 UT and leaving the imager FOV at 12:07 UT (where "North" here is defined as being along 0° longitude by convention). Figure 2, for August 18, 2004, shows a gravity wave propagating south at 157° with $\lambda_h$ = 16 km and $T_{obs}$ = 8 min, first appearing at 21:54 UT and departing from the imager FOV at 22:32 UT.

We then proceeded to perform an initial series of ray-tracing runs using these two waves. Our goal was two-fold: first, as a

proof of concept for the application of the ray-tracing model to waves in the polar MLT, and second to demonstrate the need to run the model on an atmospheric background with synoptic-scale variation. Following this, we performed ray-tracing model runs on the remainder of the gravity waves in the dataset.

## 3 Gravity wave source determination using the GROGRAT ray-tracing model

Ray-tracing techniques have been applied for decades in modeling the propagation of waves through the atmosphere (Lighthill,

1978). Dunkerton and Butchart (1984) used a simple hydrostatic ray tracing scheme to show that meridional asymmetry in the background flow due to a sudden stratospheric warming led to regions through which stationary gravity waves with horizontal wavelengths between 50-200 km could not propagate due to critical level filtering. The development of a full, three dimensional nonhydrostatic (i.e. one in which $\frac{\partial p'}{\partial z} + \rho g \neq 0$) ray tracing algorithm by Marks and Eckermann (1995), and their subsequent



additions in Eckermann and Marks (1997) led to the Gravity Wave Regional or Global Tracer (GROGRAT) ray tracing model. The model tracks the amplitude evolution and four dimensional propagation of a wave through a background atmosphere and includes terms for radiative dissipation, amplitude saturation, and turbulent diffusion, with an upper altitude limit of 120-km. The model utilizes an internal regridding scheme that permits the use of practically any input background atmosphere, allowing

for the incorporation of multiple atmospheric data products into a single run regardless of their original grid.

   GROGRAT has been used in a number of studies of wave propagation, both running in reverse for the purpose of determining tropospheric wave sources (Gerrard et al., 2004; Brown et al., 2004; Vadas et al., 2009), and for forward modeling (Lin and Zhang, 2008; Yamashita et al., 2013) the ray propagation from baroclinic regions or during disturbed conditions, such as during sudden stratospheric warmings. Ray-tracing analysis has previously been applied to the high latitude MLT by Yamashita et al.

(2013) in their study of gravity wave propagation during sudden stratospheric warming events, albeit with an arbitrary spectrum of waves originating in the troposphere and propagating into the middle atmosphere under varying background conditions. For our analysis of wave sources over SPA, we also utilized GROGRAT v2.9, with a grid displaced 4° latitude from SPA. This avoid complications around the pole arising from the singularity at -90° latitude. We ran the model on a global 2.5° x 2.5° spatial grid with 50 altitude levels spaced 2 km apart centered over the SPA site.

An important consideration in applying reverse ray-tracing techniques to gravity wave propagation through the atmosphere is the construction of an accurate atmospheric background through which the wave ray path is integrated. Two options were investigated and are presented in example runs for the waves shown in Figures 1 and 2. The first is a purely "climatological" atmosphere and the second is an atmosphere that incorporates synoptic variation below 50-km. "Climatological" runs used a background atmosphere constructed from the Navy Research Laboratory Mass Spectrometer and Incoherent Scatter Radar

(NRLMSISE-00) (Picone et al., 2002) empirical atmospheric model and the Horizontal Wind Model (HWM-93), an empirical horizontal neutral wind model of the upper atmosphere (Hedin et al., 1996), for the entire atmosphere from the surface to 120-km altitude. "Synoptic" runs utilized the European Centre for Medium-Range Weather Forecasts (ECMWF) Tropical Ocean and Global Atmospere (TOGA) (European Centre for Medium-Range Weather Forecasts, 1990) 2.5° Global Surface and Upper Air Analysis datasets below 50-km, with NRLMSISE-00 and HWM-93 input from 50-km to 100-km. Gravity

waves were initiated at 95 km with prescribed spatial and temporal characteristics as determined by our analysis of the all-sky imager data. The results for the wave observed on August 6, 2004 are shown in Figures 3a and 3b for the climatological and synoptic runs respectively. Those for August 18, 2004 are shown in Figures 4a and 4b for the climatological and synoptic runs respectively.

   For the August 6th wave, both types of runs show gravity wave rays terminating in the troposphere, at 7 km altitude for the

climatological run and at the surface for the synoptic run. However, the ray paths for the two model runs differ significantly in both direction of propagation and distance from SPA. During this period, the polar vortex, through which the wave propagates, is fairly stable as seen in the NRLMSISE-00 background in Figure 3a, while the shape of the vortex seen in the ECMWF background in 3b is distorted by apparent interaction with a planetary wave.

   A different result is seen for the wave observed on August 18. The climatological run once again produces a ray path stopping

in the troposphere near SPA at an altitude of 7 km. In the ECMWF-based synoptic model run the ray path travels down into



the stratosphere, where it travels farther out than for the climatological run, before stopping at a height of 42.5-km roughly 3.5° latitude from SPA. The polar vortex is displaced away from its normal configuration centered close to SPA and tilted in the region where the wave is determined to originate. This can be seen more clearly in the 3-dimensional projection shown in Figure 5.

5    All 94 wave events were ray-traced using GROGRAT. Seven waves were found to be evanescent, indicating they are not propagating gravity waves and are likely to be observations of local convective or dynamical instability processes in the mesopause over SPA. Figure 6 shows plots comparing the source region heights with observed wave parameters for the remaining 87, freely propagating, waves. 41 of the gravity waves were traced to tropospheric sources, while 16 waves originated above 50-km. As ECMWF does not extend beyond 50-km altitude, we were unable to analyze the sources of these waves. As shown in Figure 6, there is no correlation between the height of the wave sources and the spatial and temporal characteristics of the waves. Of the 30 remaining waves, 15 were traced into the tropopause between 9 km and 15 km and in the stratosphere between 15 km and 50 km. Based on our results the gravity waves above SPA appear to originate in several discrete layers centered at 65 km, 40 km, the tropopause, and the surface. All but 6 of the waves originated within 2.5° latitude of SPA, as seen in the bottom right panel of Figure 6, which shows the distrubution of the 87 freely propagating waves around SPA.

## 4   Analysis of Background Source Conditions using ECMWF Reanalysis

In order to identify possible wave generating regions for our the observed waves and modeled wave sources, we examined the background atmospheric conditions around SPA, within the limitations of available data products for the Antarctic lower and middle atmosphere. For this investigation we analyzed 24-hour time-differenced geopotential heights and temperatures obtained from ECMWF Reanalysis from the surface up to 50-km, the upper limit on ECMWF. We mapped 24-hour differenced geopotential heights and temperatures along the wave ray paths as determined by the GROGRAT model runs, as well as in the longitudinal direction opposite from the wave's ray path, such that each slice of data corresponded to a single longitude bin between 0-50 km altitude and -70° to -70° latitude. By examining 24-hour variations, we are able to see shifts in the structure of the polar vortex towards configurations of high baroclinicity that we would not otherwise be able to as easily infer from the raw geopotential height and temperature maps. Then, by comparing these differenced maps to the wave ray paths we can determine if wave sources match regions where baroclinic instabilities or other observable wave source regions are likely to occur.

Figure 7 shows 24-hour time differenced ECMWF geopotential height and temperature analyes of waves that were found to form in the stratosphere from July 18, 2003, July 22, 2003, August 2, 2003, and August 18, 2004, in regions where the differenced geopotential height maps are heavily slanted latitudinally and vertically, indicating a displacement of the polar vortex that has moved the polar vortex "off-balance" and has likely set up the baroclinic instability that is driving wave excitation. At mid-latitudes a westward tilt is required for a baroclinic wave to draw potential energy from the westerly mean flow (Holton, 1982), but at polar latitudes any displacement from the mean configuration centered over pole is seen as a generator of gravity waves. Our analysis is further complicated by the lower number of latitude bins near the pole, particularly when one considers



that the majority of observed wave sources come from within 2.5° of SPA. While the direction of tilt can vary latitudinally either towards or away from the pole, this does not appear to affect the formation of the waves, though this may affect the direction of horizontal wave propagation, which would become apparent in a more thorough study over an extended period.

Plots for waves observed on July 19, August 3 and August 17, 2003, and August 9, 2004 are shown in Figure 8. These waves

form in the tropopause in regions of disturbed geopotentials and temperatures. The signature of a planetary wave is present in each case in the vicinity of the wave source, which is the likely cause of the vertical forcing that is generating the waves over SPA. This structure is found in all 15 cases of waves generated in the tropopause.

## 5   Discussions and conclusions

Our observations and model analyses demonstrate that any displacement of the polar vortex, whether locally in the tropopause

due to the planetary wave interaction or as a whole in the stratosphere, is sufficient to generate upward propagating, and thus upward momentum transporting, gravity waves above the troposphere. However, several questions and concerns still remain. We are limited in terms of the available dataset both due to repeated >7 day long gaps for which no Na airglow data is available as well as the near constant presence of auroral contamination in the filter for all UT except the early morning. While there are other days available for the 2003-2005 austral winters, as previously analyzed by Suzuki et al. (2011), these are largely

disparate and spread out with larger gaps for which no Na data is available, and thus we have ignored these for now, focusing on periods of continuous observation over ∼7 day intervals.

Due to the rapidly changing background atmospheric conditions responsible for gravity wave excitation, and our reliance on NRLMSISE-00 and HWM-93 climatologies above 50 km, we are able to analyze the results of the ray tracing runs with ray paths terminating in the mesosphere to only a limited extent. Two examples of this are runs for August 6th and 7th, 2004, where

the wave rays originated at 65 km. Differenced geopotential and temperature plots for these two cases are shown in Figure 9. As the polar vortex extends upward into the MLT, the apparent disturbance of the polar vortex below 50 km seen in both figures should similarly extend upward, and is likely to be the source of the waves we observed over the SPA site. However, without the availability of a model that can account for synoptic-scale variation for the polar mesosphere for this time period we are unable to further our analysis. This is unfortunate, as waves in this region account for 16 of the 87 waves found by our model

to be real, propagating waves, and this is roughly equal in number to the waves originating from the stratosphere or tropopause.

Another consideration is our current reliance on model winds for the characterization of gravity wave intrinsic frequencies and vertical wave numbers, both necessary components as inputs into GROGRAT. Any divergence of the real background winds from the model represents a source of error for our model runs, though with winds typically being low near the pole during winter this is not expected to be a large error source. While a real vertical wind profile over SPA would be ideal, the

inclusion of available meteor radar winds at 95-km could resolve this problem.

In this paper, we have shown through the combination of observation and numerical modeling that the polar tropopause and stratosphere is a frequent source of upward propagating gravity waves. While there are inherent limitations to our analysis both



in terms of available image and atmospheric data and in refining our modeling efforts with additional, existing data, we have presented a compelling case for a previously unidentified source of small-scale gravity waves in the polar MLT.

Previous analyses of the Arctic polar vortex by Bhattacharya and Gerrard (2010) have looked at the response of the polar vortex during quiet conditions to drivers in the MLT as a form of downward control by thermospheric winds. These winds are

5 known to, in turn, respond to variations in gravity wave input into the region. With both upward and downward energy transport affecting dynamics throughout the lower and middle atmosphere, we are left with an extensive coupled system with built-in feedback mechanisms. The excitation of gravity waves in the tropopause and stratosphere by the establishment of baroclinic instabilities through displacement of the polar vortex is an important component in the system in need of further study.

*Acknowledgements.* This study was supported by the National Science Foundation with grants PLR-1247975 and 942 PLR-1443507 to the

10 New Jersey Institute of Technology. Access to SPA imager data can be obtained from http://www.antarcticgeospace.org. ECMWF TOGA Analyses can be otained from the National Center for Atmospheric Research (NCAR) Research Data Archives (RDA) from http://rda.ucar.edu/datasets/ds1 NRLMSISE-00 can be obtained from http://ccmc.gsfc.nasa.gov/modelweb/models/nrlmsise00.php, and the horizontal wind model is available at ftp://hanna.ccmc.gsfc.nasa.gov/pub/modelweb/atmospheric/hwm93/, both from the NASA Community Coordinated Modeling Center (CCMC).



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





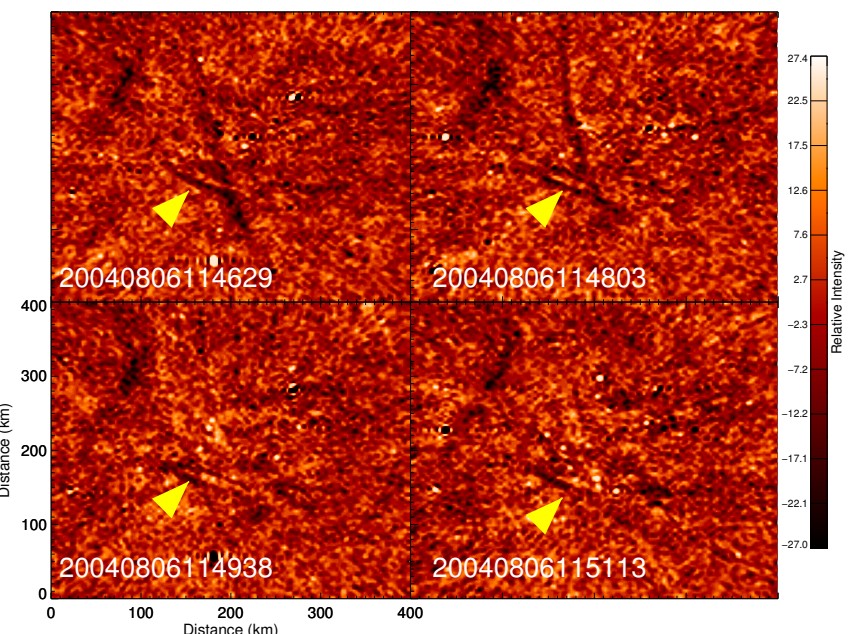

**Figure 1.** Processed Na image from August 6, 2004. The images were unwarped onto a 400 x 400 km geographic grid (shown in the bottom left image) with the positive y-axis corresponding to 0° longitude. Yellow arrows mark the location of the observed wave in each image. Time stamps are shown in the bottom left of each image, and is read as YYYYMMDDHHMMSS. The sequence of images starts at the top left, and follows to the top right, bottom left, and finally bottom right.





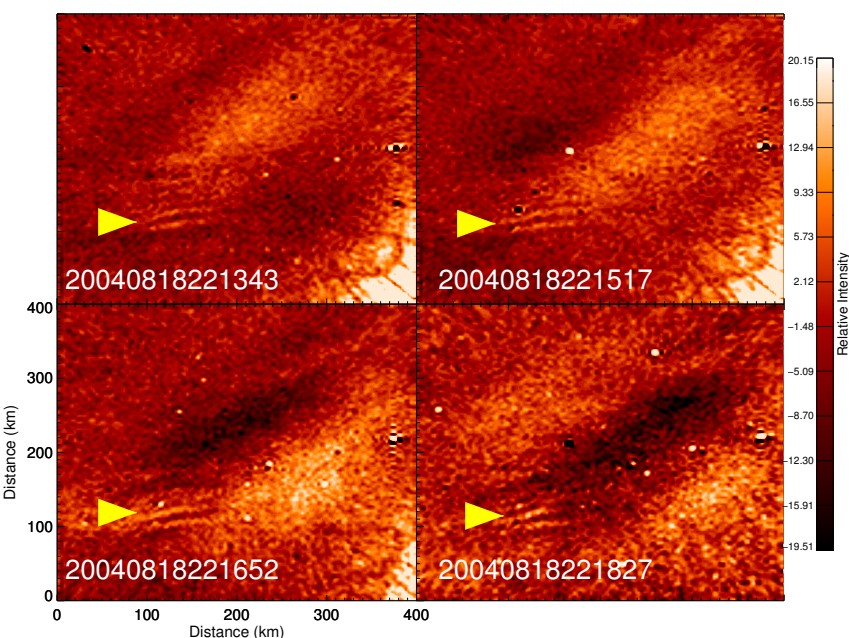

**Figure 2.** Processed Na image from August 18, 2004. The images were unwarped onto a 400 x 400 km geographic grid (shown in the bottom left image) with the positive y-axis corresponding to 0° longitude. Yellow arrows mark the location of the observed wave in each image. Time stamps are shown in the bottom left of each image, and is read as YYYYMMDDHHMMSS. The sequence of images starts at the top left, and follows to the top right, bottom left, and finally bottom right.





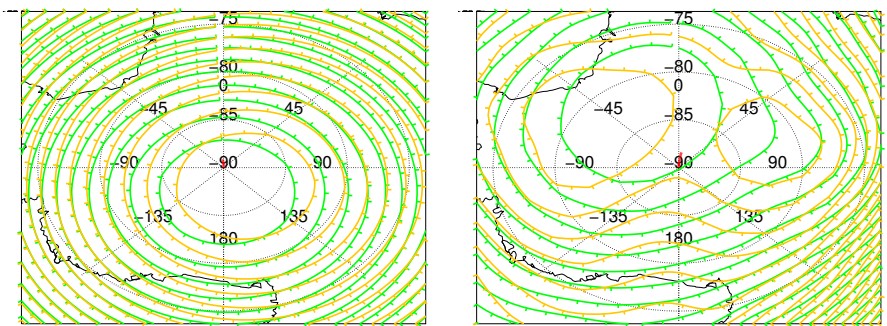

**Figure 3.** (left) Results of the GROGRAT "climatological" run for the wave observed on August 06, 2004 using background pressures, temperatures, and horizontal winds reconstructed from NRLMSISE-00 and HWM-93. (right) Results of the GROGRAT run for the same wave using an atmosphere constructed from ECMWF Reanalysis below 50 km altitude and NRLMSISE-00 and HWM-93 between 50 km and 100 km altitude. The two contours in each panel represent geopotential heights at 3 mbar (orange) and 10 mbar (green), and the red line in each panel represents the wave ray path.





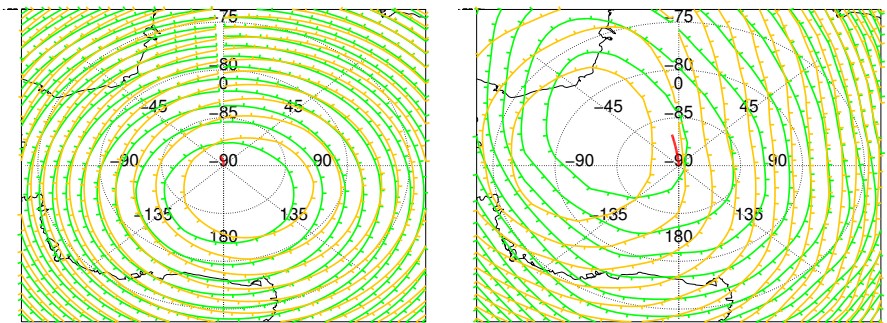

**Figure 4.** (left) Results of the GROGRAT "climatological" run for the wave observed on August 18, 2004 using background pressures, temperatures, and horizontal winds reconstructed from NRLMSISE-00 and HWM-93. (right) Results of the GROGRAT run for the same wave using an atmosphere constructed from ECMWF Reanalysis below 50 km altitude and NRLMSISE-00 and HWM-93 between 50 km and 100 km altitude. The two contours in each panel represent geopotential heights at 3 mbar (orange) and 10 mbar (green), and the red line in each panel represents the wave ray path.

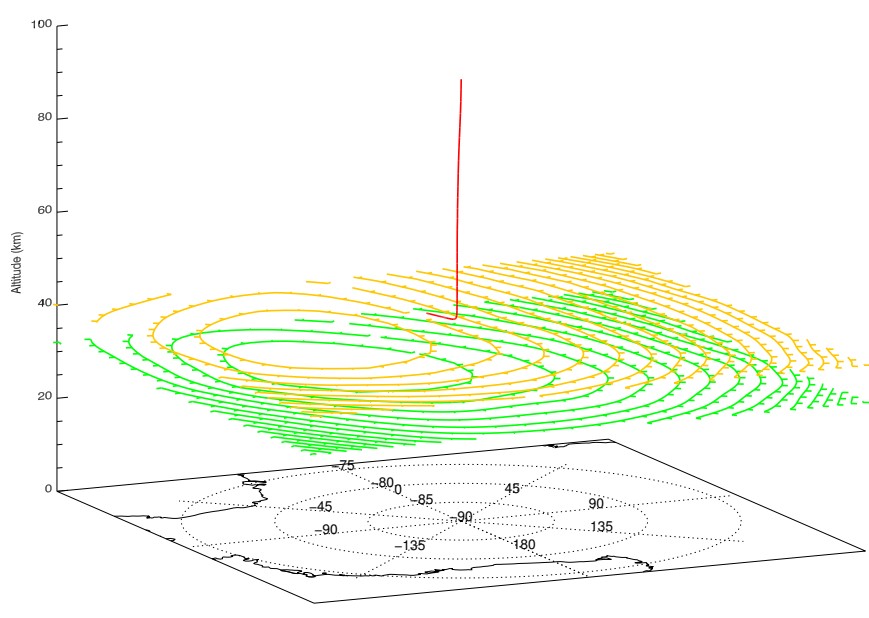

**Figure 5.** GROGRAT ray-tracing results for the August 19, 2004 wave projected in 3-D over Antarctica. The two contours represent geopotential heights at 3 mbar (orange) and 10 mbar (green), and show the wave ray path (red line).





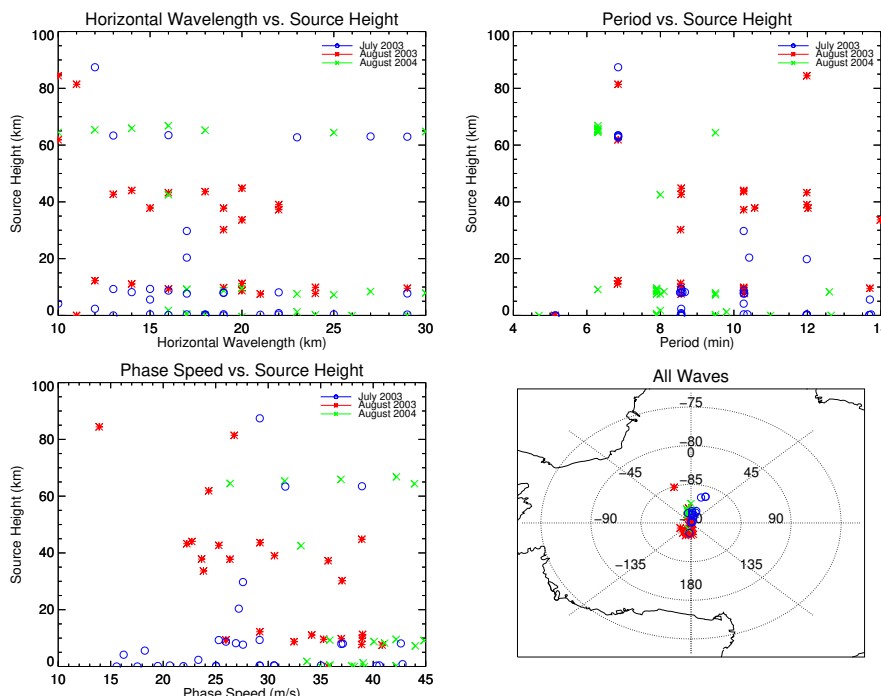

**Figure 6.** Plots comparing horizontal wavelength(top left), period(top right), and phase speed (bottom left) of the observed waves to the height of their sources as determined by individual GROGRAT runs for the 87 wave events found to be freely propagating waves. The waves are differentiated by month and year, with blue circles representing waves observed during June 2003, red 'x' marks denoting waves observed during August 2003, and green 'x' marks showing waves observed during August 2004. The bottom right panel shows a plot of the latitude and longitude of the wave sources near South Pole, from which it is apparent that all but 6 waves originate within 2.5° of SPA





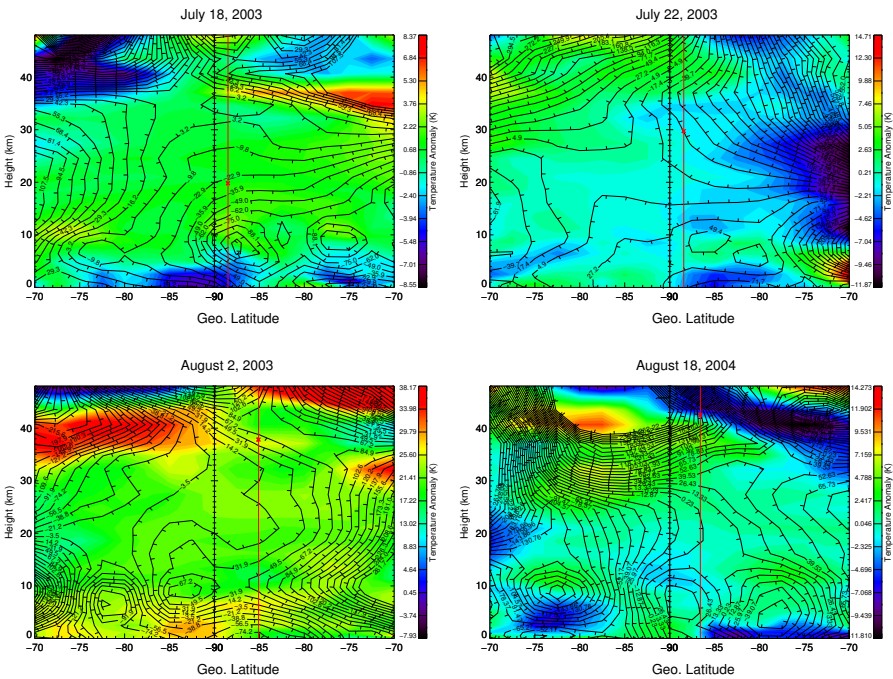

**Figure 7.** 24-hr time differenced contour plots of geopotential height (black contours) and temperatures obtained from ECMWF Reanalysis from 0-50 km along the direction of the ray path for waves observed on July 18, 2003 (top left), July 22, 2003 (top right), August 2, 2003 (bottom left) and August 18, 2004 (bottom right), as determined by our GROGRAT model runs. Ticks on contour lines point to lower geopotential height. Vertical red lines mark the latitude at which the rays terminate, and the corresponding red 'X' denotes the location of the wave source.




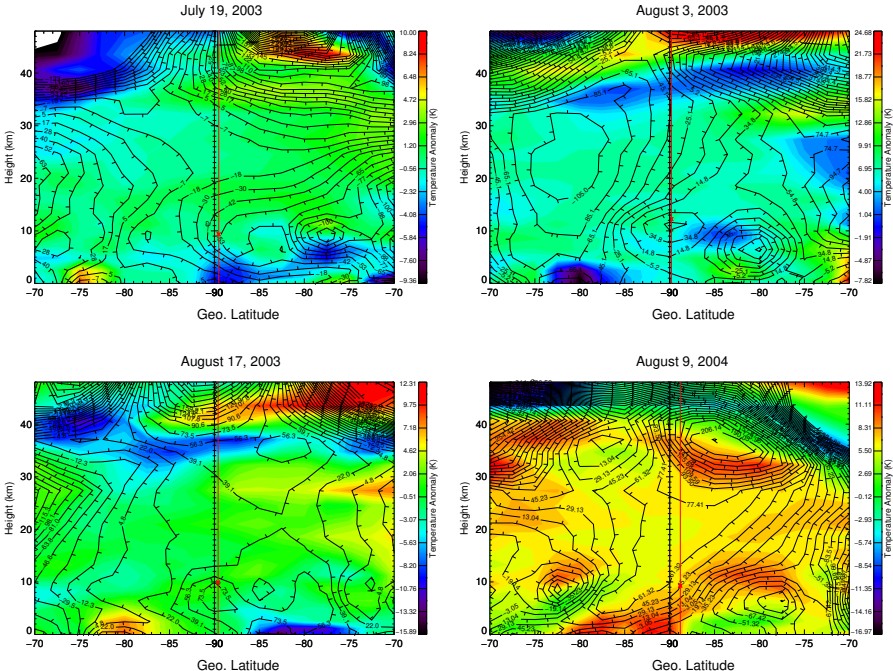

**Figure 8.** 24-hr time differenced contour plots of geopotential height (black contours) and temperatures obtained from ECMWF Reanalysis from 0-50 km along the direction of the ray path for waves observed on July 19, 2003 (top left), August 3, 2003 (top right), August 17, 2003 (bottom left) and August 9, 2004 (bottom right), as determined by our GROGRAT model runs. Ticks on contour lines point to lower geopotential height. Vertical red line marks the latitude at which the ray terminates, and the corresponding red 'X' denotes the location of the wave source.



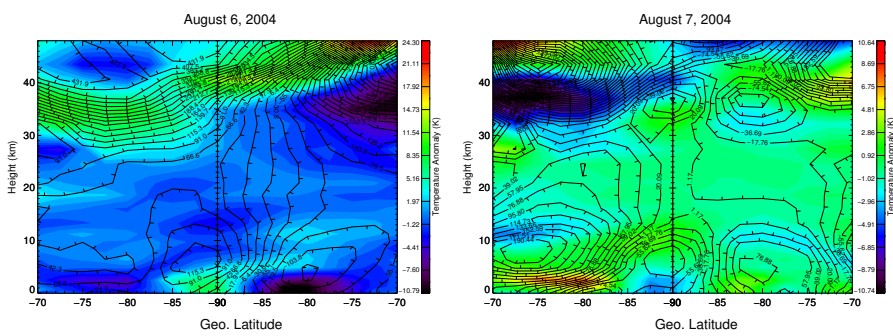

**Figure 9.** 24-hr time differenced contour plots of geopotential height (black contours) and temperatures obtained from ECMWF Reanalysis from 0-50 km along the direction of the ray path of the August 6, 2004 (left) and August 7, 2004 (right) waves, as determined by GROGRAT. Ticks on contour lines point to lower geopotential height.