# Peer review of "Short-period mesospheric gravity waves and their sources at the South Pole"

_Atmospheric Chemistry and Physics, 2016_

## Referee Comment (RC1) · Anonymous Referee #3 · 25 May 2016

Comments on "Mesospheric gravity waves and their sources at the South Pole"

The paper presents an interesting case study using data form 2003 and 2004 at SPA station. Overall I am happy with the paper, there are a couple of things I would like to see changed or added in to make it a better paper. Once these recommendations have been addressed I am happy for the paper to be published.

Minor comments:

Page 2 line 24: define NJIT

Page 5 line 5: I am assuming that the 94 events that are mentioned here use the ECMWF+NRLMSISE-00 background atmosphere rather than just the climatological background atmosphere? It needs to be clearer which background atmosphere you

are using here.

Page 6 line 29: The authors are discussing wind divergences as a source of error for their results and say "while a real vertical wind profile over SPA would be ideal, the inclusion of available meteor radar winds at 95km could resolve this problem". If they have the data already and it can help resolve how much error there could be introduced into their ray tracing then they should use it. I would like to see evidence that they have looked at the meteor winds and how they compare to the model winds around the mesopause region. I'd expect there is radiosonde data from SPA too so they would be able to get wind data for the troposphere and lower stratosphere to compare the model winds with too.

Page 7 line 2: It is not clear which of the sources they've identified they are saying in an identified one. This should be explained.

Figure 1: I find it very difficult to identify the wave fronts in this figure (Figure 2 is better). Maybe you could highlight the wave fronts rather than put and arrow in to make it easier for the reader to identify them?

Figures 3 and 4: The yellow lines are hard to make out. I'd suggest changing the yellow to something like red and then changing the red line to blue. Also, I appreciate they are showing the vortex shape but seeing the "line" in Figure 3 and 4a is difficult. Maybe they could have a zoomed in plot too showing the line more clearly?

Figure 5: This figure doesn't really convey what the authors say it should, it is quite difficult to make out the contours and the path of the wave just looks like it goes diagonal a bit the straight up. I can't see that this Figure adds anything to the paper so maybe it should be removed. I will leave this decision up to the authors.

Please also note the supplement to this comment:
http://www.atmos-chem-phys-discuss.net/acp-2016-252/acp-2016-252-RC1-supplement.pdf

---

## Referee Comment (RC2) · Dhvanit Mehta et al. · 4 Jun 2016

The paper presents a case study of mesospheric gravity waves detected in airglow
emission above the South Pole using data from three austral winter months in 2003
and 2004. The authors identify likely wave source regions based on backward ray-
traces using the GORGRAT ray-tracing model. Notably, Mehta et al. find evidence for
gravity wave sources in the lower mesosphere.

While I enjoyed reading the paper, I feel that limitations and uncertainties associated
with backward ray-tracing are not satisfactorily discussed. There are two major sources
of error which contribute to uncertainties in the computed trajectories: 1.) uncertain-
ties in the initial wave parameters (horizontal wavelength, direction of propagation,
observed period) which are derived from airglow observations in this paper, and 2.)
uncertainties in the background wind and temperature fields. Depending on the state
of the atmosphere, small changes in the direction of propagation or in the horizontal

wavelength may cause the wave's ray path to terminate at vastly different locations. The problem becomes more severe when the polar vortex is displaced and rays propagate though strong shear flows. As Mehta et al. interpret the termination point of their ray paths as potential gravity wave source regions, uncertainties in the backward trajectories may lead to a large volume with potential sources instead of single source regions. This is my major concern with this case study. The authors compare ray paths which result from using different atmospheric background fields (climatologies and ECMWF analyses). I suggest that the authors also investigate the sensitivity of the wave's ray path to variations in the initial wave parameters. It would be helpful if the authors could provide estimates of the accuracy of their derived wave parameters. For example, Figure 1 looks rather noisy and I find it difficult to motivate a propagation direction of precisely 207° (page 3, line 20). The same concerns apply to the derivation of the horizontal wavelength and observed period. I recommend the paper for publication provided the issues mentioned above are adequately addressed.

Minor comments:

Page 2, line 24: What is NJIT? Please spell out.

Page 4, line 24: The authors use ECMWF data below 50 km altitude and NRLMSISE-00 an HWM-93 above. How were the data sets stitched together? I assume there are significant differences between a climatological model and ECMWF analyses. The two data sets need to be joined somehow in order to obtain smooth background fields suitable for ray-tracing. I suggest the authors investigate how this "transition zone" affects the computed ray paths (e.g. transition at different altitudes).

Page 5, lines 2-5: "The polar vortex is displayed away from its normal configuration centered close to SPA and tilted in the region where the wave is determined to originate. This can be seen more clearly in the 3-dimensional projection shown in Figure 5." The contour lines are difficult to relate to the coordinate system in the 3D projection. I suggest a 2D plot like Figure 4.

[Figure]

Figure 5: The kink in the wave's ray path at ~43 km looks suspicious to me. The authors combine climatological winds with ECMWF analyses. I expect significant differences in the wind fields, especially when the vortex is displaced. This may introduce artificial wind shears and thus refraction of gravity waves where the two data sets are joined.

Page5, lines 5-14: I assume ECMWF data were used as background fields in the lower atmosphere (no "climatological" runs). Please clarify.

Page 6, line 28: "Low" winds at the pole during winter may help to reduce the error in estimates of intrinsic wave parameters, but even small wind speeds can cause gravity waves to be significantly refracted if the waves encounter strong shear flows. This may happen when the vortex is displaced.

Page 6, line 29: The authors mention that meteor radar winds are available at South Pole. I suggest that the authors use these data instead of the HWM-93 climatology as background winds for ray tracing or at least compare the climatology to observations (meteor radar data) in order to estimate potential errors in ray tracing.

Page 7, line 2: It is not clear to me what the authors mean by "we have presented a compelling case for a previously unidentified source of small-scale gravity waves in the polar MLT". The backward ray traces presented in this paper terminate at different altitudes in the troposphere, stratosphere and lower mesosphere.

---

## Referee Comment (RC3) · Anonymous Referee #2 · 8 Jun 2016

Review opinion on "Mesospheric gravity waves and their sources at the South Pole" by Mehta et al.

Summary:
The manuscript presents interesting analyses on the wave sources of the small-scale gravity waves observed in the winter mesosphere over South Pole. This topic is of great interest to the field of middle atmosphere research since very few studies previously focused on the generation mechanisms of such waves at Polar Regions. Utilizing GROGRAT ray-tracing model and by constructing a background atmosphere with both empirical and more "realistic" model runs, the authors located the sources for 87 wave cases observed by an all-sky imager. The results show that a remarkable number of waves (30 out of 87) are generated near the polar vortex either through baroclinic instability or interactions with planetary waves. The idea that the small-scale gravity waves (<100 km) were generated by baroclinic instability is novel yet needs more evidence and elaborated analyses. I do have a number of major comments that I would like to see the authors address before recommendation for publication.

Major comments:
1. The title does not accurately represent the research in the way that it suggests the scope of the study covers the entire wide spectrum of gravity waves that are observed in the mesosphere over South Pole. But in fact, this study is only focused on the short-period (<14 min) portion of the gravity waves. Add "short-period" in the title.

2. In the abstract, the authors mentioned "long vertical wavelength", but then there is no mentioning of vertical wavelength of these short-period gravity waves in the entire main body of the manuscript.

3. Page1, Line 19: "…, where few manned station exist to operate gravity wave instrumentation during austral winter." Some references to recent mesospheric gravity wave studies at manned station in Antarctica during winter are completely missed. These include [*Chu et al.*, 2011; *Chen et al.*, 2013, 2016; *Kaifler et al.*, 2015] for observations of mesospheric gravity waves during the austral winter in the Antarctic.

4. Page 3, Line 21: Given the sampling rate is 100 sec (~ 1.7 min), is it really possible to derive wave periods as precise as 0.1 min, as in 7.9 min? Please provide the uncertainty of the derived periods and horizontal wavelengths and a rough estimation of how much the following ray-tracing results may be affected.

5. There is meteor radar at South Pole, which provided real horizontal wind data in [*Suzuki et al.*, 2011]. What is the reason for not using the same data set for a realistic background atmosphere? Due to the critical role of a realistic atmosphere background wind play in the ray tracing, at least, it is worthwhile to validate HWM-93 with the meteor radar observation. If there were a large discrepancy between HWM-93 and the meteor radar winds, how will authors address the effect of such unrealistic atmosphere background on ray tracing. Furthermore, there must be inconsistency between HWM-93 and ECMWF at the transition region (50 km). How did the authors treat this inconsistency?

6. The identifications of baroclinic instability in Figure 7 and signature of planetary waves in Figure 8 are not clear and hard to follow in both the text and figures. Please elaborate your analysis on the part how the baroclinic instability is inferred from 24-hour differenced geopotential maps. It is also helpful to mark the related features on Figures 7 and 8.

Clarifications and technical issues
1. Page 5, Line 12: "Of the 30 remaining waves, half were traced…, and the other half"
2. Page 5, Line 27: should be "analyses".

Figures:
1. The red 'X' in Figures 7 and 8 are too small to find.

References

Chen, C., X. Chu, A. J. McDonald, S. L. Vadas, Z. Yu, W. Fong, and X. Lu (2013), Inertia-gravity waves in Antarctica: A case study using simultaneous lidar and radar measurements at McMurdo/Scott Base (77.8°S, 166.7°E), *J. Geophys. Res. Atmos.*, *118*(7), 2794–2808, doi:10.1002/jgrd.50318.

Chen, C., X. Chu, J. Zhao, B. R. Roberts, Z. Yu, W. Fong, X. Lu, and J. A. Smith (2016), Lidar observations of persistent gravity waves with periods of 3-10 h in the Antarctic middle and upper atmosphere at McMurdo (77.83°S, 166.67°E), *J. Geophys. Res. Sp. Phys.*, *121*(2), 1483–1502, doi:10.1002/2015JA022127.

Chu, X., Z. Yu, C. S. Gardner, C. Chen, and W. Fong (2011), Lidar observations of neutral Fe layers and fast gravity waves in the thermosphere (110-155 km) at McMurdo (77.8°S, 166.7°E), Antarctica, *Geophys. Res. Lett.*, *38*(23), L23807, doi:10.1029/2011GL050016.

Kaifler, B., F.-J. Lübken, J. Höffner, R. J. Morris, and T. P. Viehl (2015), Lidar observations of gravity wave activity in the middle atmosphere over Davis (69°S, 78°E), Antarctica, *J. Geophys. Res. Atmos.*, *120*(10), 4506–4521, doi:10.1002/2014JD022879.

Suzuki, S., M. Tsutsumi, S. E. Palo, Y. Ebihara, M. Taguchi, and M. Ejiri (2011), Short-period gravity waves and ripples in the South Pole mesosphere, *J. Geophys. Res.*, *116*(D19), D19109, doi:10.1029/2011JD015882.

---

## Author Comment (AC1) · 17 Aug 2016

The authors would like to thank the editor and referees for the time and effort invested in providing comments and suggestions regarding the paper. Below, we have listed the reviewer comments and addressed them, and incorporated necessary and suggested revisions into the manuscript. Reviewer comments are presented in plain text while our responses are *italicized*.

Reviewer Comments 1:

 Comments on "Mesospheric gravity waves and their sources at the South Pole"

The paper presents an interesting case study using data form 2003 and 2004 at SPA station. Overall I am happy with the paper, there are a couple of things I would like to see changed or added in to make it a better paper. Once these recommendations have been addressed I am happy for the paper to be published.

Minor comments:

Page 2 line 24: define NJIT

*We thank you for pointing out this oversight on our part. NJIT is the New Jersey Institute of Technology, the home institution of several of the authors. We have added a clarification of the acronym into the manuscript.*

Page 5 line 5: I am assuming that the 94 events that are mentioned here use the ECMWF+NRLMSISE-00 background atmosphere rather than just the climatological background atmosphere? It needs to be clearer which background atmosphere you are using here.

*Thank you for your comment. We have added a clarification that the model runs of the 94 wave events used in our study were all performed using a ECMWF+NRLMSISE00 background atmosphere.*

Page 6 line 29: The authors are discussing wind divergences as a source of error for their results and say "while a real vertical wind profile over SPA would be ideal, the inclusion of available meteor radar winds at 95km could resolve this problem". If they have the data already and it can help resolve how much error there could be introduced into their ray tracing then they should use it. I would like to see evidence that they have looked at the meteor winds and how they compare to the model winds around the mesopause region. I'd expect there is radiosonde data from SPA too so they would be able to get wind data for the troposphere and lower stratosphere to compare the model winds with too.

*Thank you for your comments. While the use of a vertical wind profile at SPA obtained from meteor radar would be ideal, personal communication with the instrument PI have indicated that such a vertical wind profile for the 2003-2004 period of study is not available, and that only*

*single point measurements at 95 km are available. While these measurements may still be useful in determining wave parameters, we ultimately decided to continue to use MSIS 90 km winds for determining wave parameters.*

Page 7 line 2: It is not clear which of the sources they've identified they are saying in an identified one. This should be explained.

*Thanks for the comments. In this case we are referring to baroclinic instability as a previously unidentified source mechanism of small-scale waves. We have added a clarification to the manuscript at this line, changing the sentence from "we have presented a compelling case for a previously unidentified source of small-scale gravity waves in the polar MLT." to "we have presented a compelling case for baroclinic instability as a previously unidentified source of small-scale gravity waves observed in the polar MLT."*

Figure 1: I find it very difficult to identify the wave fronts in this figure (Figure 2 is better). Maybe you could highlight the wave fronts rather than put and arrow in to make it easier for the reader to identify them?

*Thank you for your comments. We have replotted the figures with circles around the waves to make identifying them easier.*

Figures 3 and 4: The yellow lines are hard to make out. I'd suggest changing the yellow to something like red and then changing the red line to blue. Also, I appreciate they are showing the vortex shape but seeing the "line" in Figure 3 and 4a is difficult. Maybe they could have a zoomed in plot too showing the line more clearly?

*Thank you for the suggestions. We have revised the figures, changing the yellow lines to blue for easier readability. At the present time we feel keeping the "zoomed out" view is better to show the very small deviation from the South Pole of the purely NRLMSISE-00 model runs.*

Figure 5: This figure doesn't really convey what the authors say it should, it is quite difficult to make out the contours and the path of the wave just looks like it goes diagonal a bit the straight up. I can't see that this Figure adds anything to the paper so maybe it should be removed. I will leave this decision up to the authors.

*Thank you for your comments. The figure is meant to show a 3D projection of the plot shown in Figure 4 (right). It shows the wave ray descending from the observation site above SPA down to the stratosphere, where it bends in the presence of distorted polar vortex wind fields to a termination point which we consider to be the origin of the wave. We have added some clarifying text to the manuscript to make this clearer.*

Reviewer's Comments 2:

The paper presents a case study of mesospheric gravity waves detected in airglow emission above the South Pole using data from three austral winter months in 2003 and 2004. The authors identify likely wave source regions based on backward raytraces using the GORGRAT ray-tracing model. Notably, Mehta et al. find evidence for gravity wave sources in the lower mesosphere.

While I enjoyed reading the paper, I feel that limitations and uncertainties associated with backward ray-tracing are not satisfactorily discussed. There are two major sources of error which contribute to uncertainties in the computed trajectories: 1.) uncertainties in the initial wave parameters (horizontal wavelength, direction of propagation, observed period) which are derived from airglow observations in this paper, and 2.) uncertainties in the background wind and temperature fields. Depending on the state of the atmosphere, small changes in the direction of propagation or in the horizontal wavelength may cause the wave's ray path to terminate at vastly different locations.

The problem becomes more severe when the polar vortex is displaced and rays propagate though strong shear flows. As Mehta et al. interpret the termination point of their ray paths as potential gravity wave source regions, uncertainties in the backward trajectories may lead to a large volume with potential sources instead of single source regions. This is my major concern with this case study. The authors compare ray paths which result from using different atmospheric background fields (climatologies and ECMWF analyses). I suggest that the authors also investigate the sensitivity of the wave's ray path to variations in the initial wave parameters. It would be helpful if the authors could provide estimates of the accuracy of their derived wave parameters. For example, Figure 1 looks rather noisy and I find it difficult to motivate a propagation direction of precisely 207_ (page 3, line 20). The same concerns apply to the derivation of the horizontal wavelength and observed period. I recommend the paper for publication provided the issues mentioned above are adequately addressed.

*Thank you for your feedback. The reporting of uncertainties and potential sources of error is a major concern and consideration. As pointed out in a later response to Reviewer 3, we have revised our manuscript to include uncertainties on our wave parameter measurements obtained from the image data. We have also performed several model runs using different values within these ranges. Looking at the statistics of this sample of model runs shows a standard deviation of the longitude, latitude, and altitude of the wave ray termination point to be $4.4^o$, $2.6^o$, and 1.6 km respectively. We have added this error analysis to the manuscript.*

Minor comments:

Page 2, line 24: What is NJIT? Please spell out.

*We thank you for pointing out this oversight on our part. NJIT is the New Jersey Institute of Technology, the home institution of several of the authors. We have added a clarification of the acronym into the manuscript.*

Page 4, line 24: The authors use ECMWF data below 50 km altitude and NRLMSISE-00 an HWM-93 above. How were the data sets stitched together? I assume there are significant differences between a climatological model and ECMWF analyses. The two data sets need to be joined somehow in order to obtain smooth background fields suitable for ray-tracing. I suggest the authors investigate how this "transition zone" affects the computed ray paths (e.g. transition at different altitudes).

*Thank you for your comments. The ray-tracing model uses a cubic spline fit from the atmospheric parameters provided by both ECMWF and NRLMSISE-00 in order to construct a smoothed background atmosphere without sharp wind shears and gradients potentially arising from the boundary between the two atmospheric models. We have revised our manuscript to indicate this clearly.*

Page 5, lines 2-5: "The polar vortex is displayed away from its normal configuration centered close to SPA and tilted in the region where the wave is determined to originate. This can be seen more clearly in the 3-dimensional projection shown in Figure 5." The contour lines are difficult to relate to the coordinate system in the 3D projection. I suggest a 2D plot like Figure 4.

*Thank you for your suggestions. Figure 5 is a 3-dimensional plot of Figure 4, so replacing it with a 2D plot would be redundant. We have amended lines 2-5 to read "The polar vortex is displaced away from its normal configuration centered close to SPA and tilted in the region where the wave is determined to originate. This can be seen more clearly in the 3-dimensional projection shown in Figure 5, which is a projection of the 2D plot shown in Figure 4 (right)."*

Figure 5: The kink in the wave's ray path at _43 km looks suspicious to me. The authors combine climatological winds with ECMWF analyses. I expect significant differences in the wind fields, especially when the vortex is displaced. This may introduce artificial wind shears and thus refraction of gravity waves where the two data sets are joined.

*Thank you for the comments. While the use of two different background atmospheres can introduce artificial wind shears and wave refraction, the appearance of the "kink" at 43 km is not likely due to the interface of the two atmospheres, as this occurs at 50 km.*

Page5, lines 5-14: I assume ECMWF data were used as background fields in the lower atmosphere (no "climatological" runs). Please clarify.

*Thank you for the comments. This is correct, and we have added clarification to the manuscript.*

Page 6, line 28: "Low" winds at the pole during winter may help to reduce the error in estimates of intrinsic wave parameters, but even small wind speeds can cause gravity waves to be significantly refracted if the waves encounter strong shear flows. This may happen when the vortex is displaced.

*Thank you for your comments. While this is true, the line was meant more as a general statement on expected error in the presence of winds diverging from empirical model data, and was not meant to suggest that no errors were expected to arise from the discrepancy between real and empirical winds.*

Page 6, line 29: The authors mention that meteor radar winds are available at South Pole. I suggest that the authors use these data instead of the HWM-93 climatology as background winds for ray tracing or at least compare the climatology to observations (meteor radar data) in order to estimate potential errors in ray tracing.

*Thank you for your comments. While the use of a vertical wind profile at SPA obtained from meteor radar would be ideal, personal communication with the instrument PI have indicated that such a vertical wind profile for the 2003-2004 period of study is not available, and that only single point measurements at 95 km are available. While these measurements may still be useful in determining wave parameters, we ultimately decided to continue to use MSIS 90 km winds for determining wave parameters. We have revised the manuscript to reflect this.*

Page 7, line 2: It is not clear to me what the authors mean by "we have presented a compelling case for a previously unidentified source of small-scale gravity waves in the polar MLT". The backward ray traces presented in this paper terminate at different altitudes in the troposphere, stratosphere and lower mesosphere.

*Thank you for your response. We have amended the sentence to read "we have presented a compelling case for baroclinic instability as a previously unidentified source of small-scale gravity waves observed in the polar MLT." in order to clear up that we are referring to the initial observations of the waves in the MLT, which are generated by a previously unidentified lower altitude source, in this case, baroclinic instabilities.*

Reviewer Comments 3:

Review opinion on "Mesospheric gravity waves and their sources at the South Pole" by Mehta et al.

Summary:

The manuscript presents interesting analyses on the wave sources of the small-scale gravity waves observed in the winter mesosphere over South Pole. This topic is of great interest to the field of middle atmosphere research since very few studies previously focused on the generation mechanisms of such waves at Polar Regions. Utilizing GROGRAT ray-tracing model and by constructing a background atmosphere with both empirical and more "realistic" model runs, the authors located the sources for 87 wave cases observed by an all-sky imager. The results show that a remarkable number of waves (30 out of 87) are generated near the polar vortex either through baroclinic instability or interactions with planetary waves. The idea that the small-scale gravity waves (<100 km) were generated by baroclinic instability is novel yet needs more evidence and elaborated analyses. I do have a number of major comments that I would like to see the authors address before recommendation for publication.

Major comments:

1. The title does not accurately represent the research in the way that it suggests the scope of the study covers the entire wide spectrum of gravity waves that are observed in the mesosphere over South Pole. But in fact, this study is only focused on the short-period (<14 min) portion of the gravity waves. Add "short-period" in the title.

*Thank you for your suggestion, we have amended the manuscript title.*

2. In the abstract, the authors mentioned "long vertical wavelength", but then there is no mentioning of vertical wavelength of these short-period gravity waves in the entire main body of the manuscript.

*Thank you for your comment. We have removed mention of long vertical wavelength from the abstract.*

3. Page1, Line 19: "…, where few manned station exist to operate gravity wave instrumentation during austral winter." Some references to recent mesospheric gravity wave studies at manned station in Antarctica during winter are completely missed. These include [*Chu et al.*, 2011; *Chen et al.*, 2013, 2016; *Kaifler et al.*, 2015] for observations of mesospheric gravity waves during the austral winter in the Antarctic.

*Thank you for your comments. It was not our intention to discount the work of other studies in the Antarctic region, but to point out that this work on determining gravity wave sources had*

*not been achieved, in particular at the polar latitudes. We have added in several of the suggested references, though we have already made several references to Suzuki 2011.*

4. Page 3, Line 21: Given the sampling rate is 100 sec (~ 1.7 min), is it really possible to derive wave periods as precise as 0.1 min, as in 7.9 min? Please provide the uncertainty of the derived periods and horizontal wavelengths and a rough estimation of how much the following ray-tracing results may be affected.

*Thank you for your comments and concerns regarding error estimation. We have revised our manuscript to include estimates of the measurement error of the wave parameters for the waves in Figures 1 and 2, and discussed the variability in model results arising from these uncertainties. Typical measurement error falls within ± 1 km, ± 1 min, and ± 6º. The measured period should be reported as 8 min ± 1 min.*

5. There is meteor radar at South Pole, which provided real horizontal wind data in [*Suzuki et al.*, 2011]. What is the reason for not using the same data set for a realistic background atmosphere? Due to the critical role of a realistic atmosphere background wind play in the ray tracing, at least, it is worthwhile to validate HWM-93 with the meteor radar observation. If there were a large discrepancy between HWM-93 and the meteor radar winds, how will authors address the effect of such unrealistic atmosphere background on ray tracing. Furthermore, there must be inconsistency between HWM-93 and ECMWF at the transition region (50 km). How did the authors treat this inconsistency?

*Thank you for your comments. While the use of a vertical wind profile at SPA obtained from meteor radar would be ideal, personal communication with the instrument PI have indicated that such a vertical wind profile for the 2003-2004 period of study is not available, and that only single point measurements at 95 km are available. While these measurements may still be useful in determining wave parameters, we ultimately decided to continue to use MSIS 90 km winds for determining wave parameters.*

6. The identifications of baroclinic instability in Figure 7 and signature of planetary waves in Figure 8 are not clear and hard to follow in both the text and figures. Please elaborate your analysis on the part how the baroclinic instability is inferred from 24-hour differenced geopotential maps. It is also helpful to mark the related features on Figures 7 and 8.

*Thank you for your comments. We have marked the regions where we have inferred baroclinic instability with a yellow oval in the plots, and have included clarification in the figure caption.*

Clarifications and technical issues

1. Page 5, Line 12: "Of the 30 remaining waves, half were traced…, and the other half"

*Thank you, we have fixed this typo in the manuscript.*

2. Page 5, Line 27: should be "analyses".

*Thank you, we have fixed this typo in the manuscript.*

Figures:

1. The red 'X' in Figures 7 and 8 are too small to find.

*Thank you for your feedback, we have enlarged the red 'X's as well as marking the regions where we are inferring the formation of baroclinic instabilities with a yellow oval*

References
Chen, C., X. Chu, A. J. McDonald, S. L. Vadas, Z. Yu, W. Fong, and X. Lu (2013), Inertia-gravity waves in Antarctica: A case study using simultaneous lidar and radar measurements at McMurdo/Scott Base (77.8°S, 166.7°E), *J. Geophys. Res. Atmos.*, *118*(7), 2794–2808, doi:10.1002/jgrd.50318.
Chen, C., X. Chu, J. Zhao, B. R. Roberts, Z. Yu, W. Fong, X. Lu, and J. A. Smith (2016), Lidar observations of persistent gravity waves with periods of 3-10 h in the Antarctic middle and upper atmosphere at McMurdo (77.83°S, 166.67°E), *J. Geophys. Res. Sp. Phys.*, *121*(2), 1483–1502, doi:10.1002/2015JA022127.
Chu, X., Z. Yu, C. S. Gardner, C. Chen, and W. Fong (2011), Lidar observations of neutral Fe layers and fast gravity waves in the thermosphere (110-155 km) at McMurdo (77.8°S, 166.7°E), Antarctica, *Geophys. Res. Lett.*, *38*(23), L23807, doi:10.1029/2011GL050016.
Kaifler, B., F.-J. Lübken, J. Höffner, R. J. Morris, and T. P. Viehl (2015), Lidar observations of gravity wave activity in the middle atmosphere over Davis (69°S, 78°E), Antarctica, *J. Geophys. Res. Atmos.*, *120*(10), 4506–4521, doi:10.1002/2014JD022879.
Suzuki, S., M. Tsutsumi, S. E. Palo, Y. Ebihara, M. Taguchi, and M. Ejiri (2011), Shortperiod gravity waves and ripples in the South Pole mesosphere, *J. Geophys. Res.*, *116*(D19), D19109, doi:10.1029/2011JD015882.

Changes in the manuscript:

Title revised to "Short-period mesospheric gravity waves and their sources at the South Pole"

Revised the abstract to omit references to "long vertical wavelengths"

Page 1, line 19: added references suggested by Reviewer 3.

Page 2, line 27: added clarification of the acronym "NJIT"

Page 3, line 24 and 26: added uncertainties to the measured gravity wave parameters.

Page 4, line 28-30: added clarification on the cubic spline fit used in constructing the background atmosphere, in order to smooth out any potential artificial wind shears at the boundary between the ECMWF and NRLMSISE-00 regimes.

Page 5, line 7-9: Amended the lines to read "The polar vortex is displaced away from its normal configuration centered close to SPA and tilted in the region where the wave is determined to originate. This can be seen more clearly in the 3-dimensional projection shown in Figure 5, which is a projection of the 2D plot shown in Figure 4 (right)." in order to clear up confusion regarding Figure 5

Page 5, line 9-10: added discussion of variability in the model results arising from uncertainties in the measurement of gravity wave parameters from the image data.

Page 5, line 11-12: added clarification that the model runs for the 94 wave events were performed using a background atmosphere constructed from NRLMSISE-00 above 50 km and ECMWF reanalyses below 50 km.

Page 7, line 9-10: revised the line to read "we have presented a compelling case for baroclinic instability as a previously unidentified source of small-scale gravity waves observed in the polar MLT." providing clarification that we are referring to waves observed in the MLT and that baroclinic instability is the previously unidentified source of these small-scale waves.

Figure 1:

We have added yellow circles to better show the waves in the images, as seen below.

[Figure]

Figure 2:

We have similarly added yellow circles to this plot to better show the waves in the images, as seen below.

[Figure]

Figure 3:
Changed the yellow contours to blue for easier readability.

[Figure]

Figure 4:
Same as Figure 3.

Figure 5:
Changed contour colors to match previous to figures.

[Figure]

Figure 7:
Added yellow ovals to denote regions where we infer baroclinic instability. Amended the figure caption to reflect this.